# Examining Factors Influencing Early Paid Over-The-Top Video Streaming Market Growth: A Cross-Country Empirical Study [†]

Sangwon Lee [1], Seonmi Lee [2,*], Hyemin Joo [3] and Yoonjae Nam [4]

[1] Department of Media, College of Politics and Economics, Kyung Hee University, Seoul 02447, Korea; swlee2668@khu.ac.kr
[2] Institute of Economics and Business Research, Korea Telecom, Seoul 03155, Korea
[3] Kantar Korea, Seoul 07330, Korea; 1817min@naver.com
[4] Department of Culture, Tourism & Content, College of Hotel & Tourism Management, Kyung Hee University, Seoul 02447, Korea; ynam@khu.ac.kr
* Correspondence: infoecon@gmail.com
† This work was supported by the Kyung Hee University under grant [20181288].

**Abstract:** This study examines the factors influencing early paid Over-The-Top (OTT) video streaming market growth in 50 countries. The results of the panel data analysis suggest that Netflix's market entry, traditional pay TV market size, broadband infrastructure, and OTT platform competition contribute to the early market growth of paid OTT video streaming services, such as subscription video-on-demand (SVOD) services. The results also reveal that the traditional pay TV subscription market and the paid OTT video streaming market initially grow together in many countries. However, the findings also reveal a negative association between the market entry of Netflix and the subscription revenue growth rate of traditional pay TV services. The results of this study suggest industry and policy implications for paid OTT video streaming market growth and the sustainable development of the media industry.

**Keywords:** Over-The-Top video; OTT market growth; Netflix effect; SVOD

## 1. Introduction

The rapid diffusion of broadband technologies and continuous innovations in the media and information and communication technologies (ICT) industries have enabled consumers to access video content anytime and anywhere via multiple devices, including computers, smartphones, tablets, gaming consoles, television sets, and other equipment connected to the Internet [1]. As evidence of this trend, internet-based video services, such as Over-The-Top (OTT)—the most recent and potentially disruptive innovation in the media industry—have shown rapid growth in many countries. OTT is recognized as the next generation of media that will bring innovation and efficiency with the proliferation of digital transformation (Digital transformation can be understood as a continuous process by which enterprises adapt to or drive disruptive changes in their customers and markets by leveraging digital competencies to create new business models, products, and services [2].

The OTT video market is constantly growing. For example, it was estimated that internet video subscription revenues have increased from $12.5 billion in 2014 to $45.26 billion in 2019, including subscription video-on-demand (SVOD) and transactional video-on-demand (TVOD) services (In general, SVOD services charge subscribers a monthly fee (a flat rate) for access to premium content. TVOD services charge consumers for access to on-demand assets or live-streams on a pay-per-view basis. SVOD and TVOD services are paid OTT services.) [3] (see Figure 1). Furthermore, as of January 2020, Netflix, a leader among the global OTT service providers, had 167 million streaming subscribers [4]. Forecasters have estimated that the number of worldwide Netflix streaming service subscribers will reach nearly 237 million by 2025 [5].

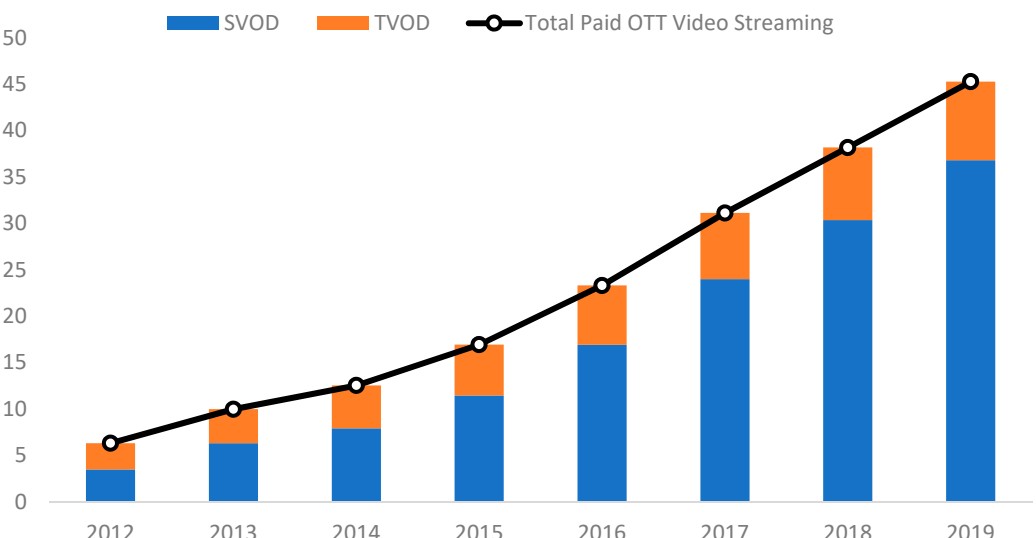

**Figure 1.** Global paid OTT video streaming market (2012–2019) (USD billions).

Notably, no absolute or globally accepted definition of OTT services seems to exist. In general, an OTT service is understood as content, a service, or an application that is provided to the end user over the open Internet [6]. In a broad sense, an OTT service includes the provision of content and applications, such as voice services provided over the internet, web-based content, search engines, hosting services, email services, instant messaging, and video and multimedia content [6].

OTT service providers, such as Skype and Netflix, are now growing at sufficient rates to compete with traditional telecommunications and broadcasting services. Consequently, any content and application provider (CAP) of online services that can potentially substitute for traditional telecommunications and broadcasting services, such as voice telephony and television, is considered an OTT player [7]. In broadcasting markets, Internet protocol TV (IPTV) and OTT service providers seem to offer similar video services in that both offer services over the internet infrastructure. However, in most cases, OTT video providers do not provide the first and last mile broadband connections to physically transmit video to consumers, while IPTV providers usually do [8]. IPTV is generally offered by telecommunications operators, such as AT&T U-Verse in the United States (US), over a managed network with a guaranteed quality of service, while OTT services are provided by content owners, such as the BBC in the United Kingdom (UK), or dedicated start-up players, such as Netflix in the US, without the involvement of internet service providers (ISP) or network operators in either the control of the content or its access by viewers [9].

In spite of the rapid growth of OTT video subscribers in global media markets, few empirical studies have employed a panel data set to examine the factors influencing early market growth of paid OTT video streaming services, such as SVOD services. In addition, only a few studies have examined the impacts of paid OTT video streaming market growth on the traditional pay TV markets [10,11]. Specifically, no empirical work has investigated whether the paid OTT video streaming market growth rate positively (or negatively) influences the pay TV subscription revenue growth rate. In addition, few empirical studies have examined whether the market entry of global OTT providers, such as Netflix, positively influences OTT market growth and negatively influences pay TV subscription revenue growth.

Employing a panel data set, this study attempts to make a significant, original contribution to the literature by examining factors influencing paid OTT video streaming market growth. Employing fixed effects regression models, this study examines factors influencing early market growth of paid OTT video streaming services, such as SVOD services, in 50 countries. A panel data analysis is adopted, using secondary data from 2012 to 2016. The

results of this empirical study suggest industry and policy implications for paid OTT video streaming market growth and the sustainable development of the media industry.

## 2. Literature Review

### 2.1. Global OTT Video Streaming Market Growth

The introduction of OTT services, such as Netflix, Amazon, Hulu, and YouTube, has revolutionized media. Offered as cheap unbundled content, OTT services bring users many benefits, which leads to cost savings and wider access to educational resources from open-source platforms [12].

Based on the diffusion of broadband network and smart devices, global markets for OTT services will continue to grow. For instance, forecasters have estimated that, thanks to an unprecedented growth spurt, fueled in part by the COVID-19 pandemic, Netflix passed the 200 million paid subscribers around the world mark in 2020 [13]. Netflix launched its innovative video service and expanded its services geared toward global markets. Of Netflix's total number of subscribers, 58% were from outside the US at the beginning of January 2019 [4]. Netflix's global streaming services are currently available in more than 200 countries around the world. As Netflix's subscriber base grows, Netflix benefits from economies of scale due to its declining average cost. Netflix's performance is also due to its investment in original content and innovative platform service. It is estimated that Netflix's video content budget in 2020 could have surpassed $17 billion [14]. Beyond Netflix, it is estimated that total global paid OTT video streaming service revenues will reach $72.75 billion in 2023 [3]. It was estimated that the top four large global and semi-global players, namely Netflix, YouTube, Hulu, and Amazon Prime, compete across multiple markets and collectively control over 40 percent of annual OTT video market revenues [3,4].

### 2.2. Complementary Goods and Substitutional Goods

In some countries (e.g., the US), OTT video streaming services have largely replaced traditional pay TV subscription services, such as cable TV services [15]. In the US, in 2017, Netflix had more subscribers than the largest cable TV [15]. Forecasters have estimated that the number of SVOD service subscribers in the US will reach twice the number of pay TV service subscribers in 2022 [16]. Thus, the substitution of pay TV services by OTT video streaming services produced a newly coined term, 'cord-cutting', which refers to the practice of canceling a cable TV subscription in favor of OTT video services.

In spite of this cord-cutting trend, global traditional pay TV subscription service markets are still growing. Forecast estimates are that global traditional pay TV subscription revenues will increase from $3.5 billion to $5.9 billion between 2012 and 2021 [17]. Based on the comparison of SVOD household penetration in 2018 and percent change in pay TV penetration between 2016 and 2018, in some European countries, including France, the UK, and Italy, SVOD has thus far been complementary to pay TV platforms [10]. Thus, the interaction between paid OTT video streaming service and pay TV service uptake is complex and varies significantly by market [10].

Previous studies in the media economics field examined the user substitutability and complementarity of diverse media. For instance, analyzing individual-level media consumption data from a media diary, Jang and Park [18] discovered significant substitution among newspaper, television, and computer use, while mobile telephone and computer use seem to be complementary regarding time of use. Lee et al. [19] revealed that tablet PCs are a complement to smartphones in the early diffusion of smart devices.

Displacement theory also involves with the substitutability of media. A key assumption of displacement theory is that, when a new communication technology is introduced, we then have less time for other communication activities. The theory assumes that time is a limited resource and inelastic [20]. Therefore, displacement theory implies that time dedicated to a medium can also supplant time saved for other important everyday activities, including other media use [21].

Therefore, considering the trend of OTT video streaming market growth, it is interesting to examine whether OTT video streaming market growth has positively or negatively influenced traditional pay TV subscription market growth. Based on the literature review, the following research question is proposed:

**RQ1:** Has the paid OTT video streaming market growth rate influenced the traditional pay TV subscription revenue growth rate?

### 2.3. Market Entry and Growth–Netflix Effect

As technological convergence provides an opportunity to develop new services, communications firms tend to be incentivized to enter new technology markets to attract greater consumer segments [22]. In the communications market, a firm sometimes enters a new market either to introduce a new revenue stream by expanding its services or to reduce the firm's costs of offering services [23,24]. In the network industry, greater entry occurs because firms can reduce their fixed costs [25].

When a new firm enters a market, incumbent firms seem to improve their products, services, or facilities in order to compete with the entrant and not cannibalize their profits [26]. That is, when an innovative firm enters a market and begins to take a large market share at a rapid pace, it seems to stimulate rivals to execute some new and different strategies as a defense, such as providing new services that can lure new consumers, which would consequently give rise to market growth. Some have called this the "catfish effect," referring to the phenomenon that a catfish placed in a tank will, by its vigorous activity, oxygenate the water and energize the other fish [27]. The catfish effect can be applied to the market that Netflix has entered, and its rivals have implemented some defensive strategies. In particular, Netflix has been evaluated as bringing an innovative service to the video market by creating content recommendation algorithms that help consumers choose what they are most likely to watch [28]. When a firm, especially one with an innovation, steps into a market, it is likely to have a positive impact on the firm's performance and market growth [29–31].

However, considering that Netflix's video streaming service has replaced traditional pay TV subscription services in some countries, it is also interesting to examine whether Netflix's market entry and growth affect traditional pay TV subscription revenue growth. For instance, in the US, it was estimated that the number of US cord-cutters who have canceled their pay TV subscriptions and who will continue without it reached 32.8% in 2018 [32]. Considering that Netflix had over 56.72 million US streaming subscribers in the second quarter of 2018 [33], it seems that many US cord-cutters are Netflix users. Based on the literature review, the following research questions are proposed:

**RQ2:** Has Netflix's market entry positively influenced paid OTT video streaming market growth?

**RQ3:** Has Netflix's market entry negatively influenced the pay TV subscription revenue growth rate?

### 2.4. Broadband Infrastructure and OTT Video Streaming Market Growth

OTT services, such as the Netflix streaming service, are provided to the end user over the open internet [6]. According to the FCC [34], an online video distributor (OVD), another name for an OTT provider, is defined as any entity that offers video content by means of the Internet or other Internet Protocol (IP)-based transmission path provided by a person or entity other than the OVD. As such, the main technical means to deliver OTT video is based on broadband platforms that provide internet access. As fixed and mobile broadband networks have evolved into next generations, their capacity has increased exponentially to deliver OTT video applications that require larger bandwidths, and OTT service providers are able to seamlessly provide real-time TV or streaming video services [35]. In general, innovative ICT services can be well-developed under these advanced broadband network conditions [36]. Empirically, Lee et al. [37] showed that broadband infrastructure is a necessary condition for IPTV diffusion, and competition

among various platforms even drives IPTV diffusion more rapidly. Considering that an increasing number of OTT users prefer their smart devices, such as smartphones and tablets, over their PC when enjoying their favorite OTT video streaming services, the mobile broadband infrastructure is essential for OTT video delivery. Thus, a mobile broadband network as well as a fixed broadband network, such as fiber and/or 3G/4G/5G mobile, will likely facilitate the rapid growth of OTT. Therefore, a well-established broadband infrastructure in a country is also an important influential factor for OTT video market growth. Based on the literature review, the following research question is proposed:

**RQ4:** Does broadband infrastructure positively influence paid OTT video market growth?

### 2.5. Market Environmental Factors–Competition, Concentration and Pay TV Market Size

Market environmental factors, such as platform competition and platform concentration, are also factors influencing market growth and performance. In general, platform competition occurs when different technologies or platforms compete to provide ICT services to end-users [38]. Currently, different types of OTT video services, such SVOD and TVOD, are available in global OTT video markets. In terms of market revenue, forecast estimates are that global SVOD revenues will increase from $3.4 billion to $27.8 billion between 2012 and 2021 [17]. It is also estimated that global TVOD revenues will increase from $2.8 billion to $8.9 billion during the same period [17]. Platform competition among these different types of OTT video services can contribute to the market growth of OTT video services. Competition among different platforms may bring diverse choices and innovations for consumers [37].

Moreover, in the video markets, different traditional pay TV platforms and OTT platforms for video programming involve platform market concentration or platform competition. Platform market concentration (or competition) refers to inter-modal market concentration (or competition) among different technologies (or platforms) in a market. If OTT video services and traditional pay TV services can be defined as a total pay TV market, then platform market concentration (or competition) between traditional pay TV and OTT, as a market environmental factor, may affect the market growth of pay TV markets. Previous studies on the diffusion of new media technologies have suggested that platform competition is a driver of new media technology diffusion and growth. For instance, platform competition among different fixed broadband technologies may lead to lower prices and the growth of broadband markets [36].

In addition, market size can be related to the growth of a new service. From a business strategy perspective, market size is an important determinant for communications firms to offer new services, because it provides an opportunity for firms to attain greater returns on their investments, grow their market share, and generate new revenue streams [39,40]. Thus, media firms may expect higher potential returns in larger, untapped markets, as well as mitigate their investment risks [40]. Seo [41] also discovered that in areas where the market size is larger, media firms are likely to provide new services. In addition, a previous study also revealed that the pay TV market size contributes to IPTV penetration [37].

Therefore, the traditional pay TV market size is closely associated with an OTT video provider's strategic decision to expand its business, which consequently expands potential OTT business. Based on the literature review, the following research questions are proposed:

**RQ5:** Does platform competition among paid OTT types influence paid OTT video streaming market growth?

**RQ6:** Does pay TV/OTT platform market concentration influence the pay TV subscription revenue growth rate?

**RQ7:** Does the pay TV market size positively influence paid OTT video streaming market growth?

## 3. Research Method

A panel data set was utilized to estimate regression models of paid OTT video streaming market growth and the market growth rate of traditional pay TV subscription services. The data for the regression analyses covered the years from 2012 to 2016. For the data analyses, a total of 250 observations from 50 countries were available. To control for the unobserved heterogeneity among countries, this study employed a fixed effects model.

### 3.1. The Paid OTT Video Streaming Market Growth Model

Employing OTT market growth models, the factors affecting OTT market growth were analyzed. The empirical model (1) demonstrates the paid OTT video streaming market growth model. In the OTT market growth model, the dependent variable ($P\_OTT_{it}$) is the paid OTT video streaming market size in country $i$ by time $t$. For the independent variables, total broadband penetration, paid OTT platform competition, Netflix market entry, and traditional pay TV market size were included in the empirical model. In addition, population density and income were included in the model as control variables. Considering that SVOD services are growing rapidly in global paid OTT video streaming markets, employing the same independent variables, the factors affecting SVOD market growth were also analyzed.

To analyze the impacts of Netflix's market entry in previous periods, one and two year time-lag Netflix market entry variables were included in the second model (2). The empirical model (2) demonstrates the paid OTT market growth model, in which one- and two-year time-lag Netflix market entry variables were included. In the second paid OTT video streaming market growth model, the dependent variable ($P\_OTT_{it}$) is the paid OTT video streaming market size in country $i$ by time $t$. For the independent variables, total broadband penetration, paid OTT platform competition, Netflix market entry ($t$-1), Netflix market entry ($t$-2), traditional pay TV market size, and income were included in the empirical model.

In considering heteroscedasticity issues among countries, this study used the fixed effects model. While the pooled Ordinary Least Square (OLS) regression method assumes homogeneous firm characteristics among variables, and consequently, is likely to suffer from heterogeneity bias of coefficient estimates, the fixed effects model allows firm differences among variables, which may control for unobserved firm heterogeneity in the model. Thus, the fixed effect model helps obtain unbiased estimates compared to pooled OLS [42]. In the OTT market growth model, $\beta_0$ is constant, the $\alpha_i$ variable denotes country-specific dummy variables, and $\varepsilon_{it}$ is the error term.

$$P\_OTT_{it} = \beta_0 + \beta_1 * BROADBAND_{it} + \beta_2 * OTT\_PCOM_{it} + \beta_3 * POD_{it} + \beta_4 * Neflix_{it} + \beta_5 * PayTVMARKET_{it} + \beta_6 * INC_{it} + \gamma_i \alpha_i + \varepsilon_{it} \quad (1)$$

$$P\_OTT_{it} = \beta_0 + \beta_1 * BROADBAND_{it} + \beta_2 * OTT\_PCOM_{it} + \beta_3 * POD_{it} + \beta_4 * Neflix_{it-1} + \beta_5 * Neflix_{it-2} + \beta_6 * PayTVMARKET_{it} + \beta_7 * INC_{it} + \gamma_i \alpha_i + \varepsilon_{it} \quad (2)$$

### 3.2. Traditional Pay TV Subscription Revenue Growth Rate Model

Whether diverse OTT-related factors affect the pay TV subscription revenue growth rate was also analyzed using the traditional pay TV subscription revenue growth rate model. The empirical model (3) demonstrates the pay TV subscription revenue growth rate model. In the empirical model, the dependent variable ($PayTVCAGR_{it}$) is the traditional pay TV subscription revenue growth rate in country $i$ by time $t$. For independent variables, paid OTT revenue growth rate, Netflix market entry, pay TV_OTT HHI (pay TV/OTT platform concentration), and fixed broadband penetration were included in the empirical model. In addition, income and population density were included in the model as control variables. The traditional pay TV subscription revenue growth rate model also utilizes a fixed effects model, which controls for the unobserved heterogeneity among countries. In the empirical model, $\beta_0$ is constant, the $\alpha_i$ variable denotes country-specific dummy variables, and $\varepsilon_{it}$ is the error term.

$$LnPayTVCAGR_{it} = \beta_0 + \beta_1 * LnP\_OTTCAGR_{it} + \beta_2 * Netflix_{it} + \beta_3 * PayTV\_OTT\_HHI_{it} + \beta_4 * INC_{it} + \beta_5 * POD_{it} + \beta_6 * Fixed\_B_{it} + \gamma_i \alpha_i + \varepsilon_{it} \quad (3)$$

### 3.3. Measurement and Data Sources

Table 1 provides the variables, their measurement, and the data sources for this study. The paid OTT video streaming market size was measured by the OTT revenue (USD) per 100 inhabitants. In addition, the SVOD revenue share was measured by the SVOD revenue (USD) per 100 inhabitants. The pay TV subscription revenue growth rate was measured by the compound annual growth rate of traditional pay TV subscription revenue. As detailed in the literature review, various independent variables may affect OTT market growth. Total broadband infrastructure was measured by the total number of broadband subscribers per 100 inhabitants, including fixed and mobile broadband subscribers. In addition, fixed broadband infrastructure was measured by the number of fixed broadband subscribers per 100 inhabitants.

**Table 1.** Description of variables.

| Variables | Measurement | Data Sources |
|---|---|---|
| Paid OTT video streaming market size | Paid OTT video streaming revenue (USD) per 100 inhabitants | PwC |
| SVOD market size | SVOD revenue (USD) per 100 inhabitants | PwC |
| Pay TV subscription revenue growth rate | Compound annual growth rate of traditional pay TV subscription revenue | PwC |
| Total broadband infrastructure | Total number of broadband subscribers per 100 inhabitants (including fixed and mobile broadband) | ITU |
| Fixed broadband infrastructure | Number of fixed broadband subscribers per 100 inhabitants | ITU |
| OTT platform competition | Herfindahl–Hirschman Index for different paid OTT types | PwC |
| Population density | Population per $km^2$ | World Bank |
| Netflix market entry | Netflix market entry dummy (0 or 1) | PwC |
| Traditional Pay TV market size | Traditional pay TV subscription revenue (USD) per 100 inhabitants | PwC |
| Income | GDP per capita | World Bank |
| Paid OTT video streaming revenue growth rate | Compound annual growth rate of paid OTT video streaming revenue | PwC |
| PayTV/OTT platform market concentration | Herfindahl–Hirschman Index for traditional pay TV and paid OTT platforms | PwC |

For the measurement of OTT platform competition, the Herfindahl–Hirschman Index (HHI) for different paid OTT types, such as SVOD and TVOD, was employed. The HHI has been widely used in previous empirical studies to measure competition and market concentration in the ICT industry [43,44]. Population density was measured by population per $km^2$. For the measurement of Netflix market entry, a Netflix market entry dummy was employed (0 or 1). In addition, the traditional pay TV market size was measured by the pay TV subscription revenue (USD) per 100 inhabitants. For the measurement of income, GDP per capita was employed. For the measurement of paid OTT video streaming revenue growth rate, the compound annual growth rate of paid OTT video streaming revenue was utilized. In addition, for the measurement of OTT/pay TV platform market concentration, the HHI for traditional pay TV and paid OTT platforms was employed.

The data utilized were collected from different sources depending on the variables. Data were collected from the PwC, ITU, and World Bank Group. The 50 countries represented in the panel dataset are listed in Table 2.

**Table 2.** Countries represented in panel data set.

| 1 | Argentina | 26 | Korea (R.O.K.) |
|---|---|---|---|
| 2 | Australia | 27 | Malaysia |
| 3 | Austria | 28 | Mexico |
| 4 | Belgium | 29 | Netherlands |
| 5 | Brazil | 30 | New Zealand |
| 6 | Canada | 31 | Nigeria |
| 7 | Chile | 32 | Norway |
| 8 | China | 33 | Pakistan |
| 9 | Colombia | 34 | Peru |
| 10 | Czech Republic | 35 | Philippines |
| 11 | Denmark | 36 | Poland |
| 12 | Egypt | 37 | Portugal |
| 13 | Finland | 38 | Romania |
| 14 | France | 39 | Russia |
| 15 | Germany | 40 | Saudi Arabia |
| 16 | Greece | 41 | Singapore |
| 17 | Hong Kong | 42 | South Africa |
| 18 | Hungary | 43 | Spain |
| 19 | India | 44 | Sweden |
| 20 | Indonesia | 45 | Switzerland |
| 21 | Ireland | 46 | Thailand |
| 22 | Israel | 47 | Turkey |
| 23 | Italy | 48 | United Arab Emirates |
| 24 | Japan | 49 | United Kingdom |
| 25 | Kenya | 50 | United States |

## 4. Results

Table 3 presents descriptive statistics of the variables used in this study. For the OTT video streaming market growth models, 50 countries' data between 2012 and 2016 were analyzed. For the paid OTT video streaming market growth model, the mean of the OTT video streaming market size per 100 inhabitants was 408.35 with a standard deviation of 713.10. For the SVOD market growth model, the mean of the SVOD market size per 100 inhabitants was 270.06 with a standard deviation of 599.36. For the pay TV subscription market growth rate model, the mean of the pay TV subscription market growth rate was 0.15 with a standard deviation of 0.11.

**Table 3.** Summary statistics of key variables.

| | N | Mean | Min | Max | S.D. |
|---|---|---|---|---|---|
| Paid OTT video streaming market size | 250 | 408.35 | 0.02 | 3693.59 | 713.10 |
| SVOD market size | 250 | 270.06 | 0.01 | 2906.23 | 559.36 |
| Pay TV subscription revenue growth rate | 235 | 0.15 | −0.44 | 1.00 | 0.11 |
| Total broadband infrastructure | 249 | 87.46 | 1.18 | 183.42 | 44.94 |
| OTT platform competition | 248 | 6864.06 | 5000.00 | 10,000.00 | 1565.80 |
| Population density | 250 | 425.88 | 2.96 | 7908.72 | 1415.73 |
| Netflix market entry | 250 | 0.50 | 0.00 | 1.00 | 0.50 |
| Traditional Pay TV market size | 250 | 6433.10 | 20.04 | 31,585.74 | 6866.39 |
| Income | 250 | 31,846.86 | 2650.44 | 87,832.59 | 18,500.06 |
| Paid OTT video streaming revenue growth rate | 229 | 1.57 | −0.64 | 34.00 | 3.64 |
| PayTV/OTT platform market concentration | 250 | 9185.79 | 6904.87 | 10,000.00 | 741.06 |
| Fixed broadband infrastructure | 250 | 21.56 | 0.01 | 45.13 | 13.11 |

Tables 4–6 illustrate the correlation matrix and variance inflation factor (VIF) values of explanatory variables in respective order to see multicollinearity. As seen in Table 4, applying the 0.80 standard for the strength of the correlations, no variables were excluded for the OTT market growth model and the SVOD market growth model. These results

are also supported in checking the variance inflation factors (VIFs) in Table 5. Any value that is not above 10 is supposed to be regarded as a multicollinearity benchmark. Thus, all variables suggested in the regression model in the previous chapter (i.e., total broadband infrastructure, paid OTT market competition, population density, Netflix market entry, traditional pay TV market size, and income) were successfully utilized for the analyses. For the dependent variables, the paid OTT video streaming market size and SVOD market size, and one exploratory variable, total broadband infrastructure, were transformed using a logarithmic function because they were skewed. The rest of the variables were used without any transformation process. Moreover, no variable was excluded for the pay TV subscription market growth rate model. This result is also supported in checking the VIF values of variables in Table 6. Consequently, all variables suggested in the previous regression model (i.e., OTT revenue growth rate, Netflix market entry, PayTV/OTT platform market concentration, population density, fixed broadband infrastructure, and income) were employed for the analysis. For the dependent variable, pay TV subscription revenue growth rate and two exploratory variables, paid OTT video streaming revenue growth rate and fixed broadband infrastructure, were transformed using a logarithmic function because they were skewed. The rest of the variables were input without any transformation process.

**Table 4.** Correlation matrix.

| | Total Broadband Infrastructure | OTT Platform Competition | Population Density | Netflix Market Entry | Traditional Pay TV Market Size | Income | Paid OTT Revenue Growth Rate | PayTV/OTT Platform Market Concentration |
|---|---|---|---|---|---|---|---|---|
| OTT platform Competition | −0.2681 | | | | | | | |
| Population Density | 0.2900 | −0.0174 | | | | | | |
| Netflix market Entry | 0.2944 | −0.2157 | −0.128 | | | | | |
| Traditional pay TV market size | 0.5756 | −0.1414 | −0.0228 | 0.3981 | | | | |
| Income | 0.7948 | −0.1760 | 0.4171 | 0.1841 | 0.6015 | | | |
| Paid OTT revenue growth rate | −0.1611 | 0.2308 | −0.0228 | −0.1100 | −0.1132 | −0.0630 | | |
| PayTV/OTT platform market concentration | −0.6120 | 0.2378 | −0.2479 | −0.3566 | −0.2869 | −0.5556 | 0.1334 | |
| Fixed broadband infrastructure | 0.7684 | −0.2154 | 0.1483 | 0.2432 | 0.6798 | 0.7254 | −0.1098 | −0.4626 |

**Table 5.** Variance inflation factors: paid OTT video streaming and SVOD market growth models.

| Variables | VIF |
|---|---|
| Income | 3.66 |
| Total broadband infrastructure | 3.10 |
| Traditional pay TV market size | 1.95 |
| Population density | 1.48 |
| Netflix market entry | 1.25 |
| OTT platform competition | 1.06 |
| Mean VIF | 2.08 |

**Table 6.** Variance inflation factors: traditional pay TV subscription revenue growth rate model.

| Variables | VIF |
|---|---|
| Income | 2.43 |
| Fixed broadband infrastructure | 2.10 |
| PayTV/OTT platform market concentration | 1.57 |
| Netflix market entry | 1.15 |
| Paid OTT revenue growth rate | 1.03 |
| Mean VIF | 1.66 |

In considering heteroscedasticity issues among the 50 countries, this study used the fixed effects model. In addition, to check the continuity of the Netflix entry effect, this study considered one-year and two-year time-lag models. For the paid OTT market growth model and the SVOD market growth model, the total number of observations was 247. For the pay TV subscription revenue growth model, the total number of observations was 203.

Table 7 shows the results of the regression analysis of the paid OTT video streaming market growth model, controlling for country fixed effects. The F statistics in all regression models (both the no-time-lag model of paid OTT growth and the time-lag model) were statistically significant at the 5% level ($F(6,150) = 43.03$, $F(7,143) = 34.39$, respectively), which means that one of the regression coefficients would be different from zero. In the no-time-lag model, total broadband infrastructure, Netflix market entry, traditional pay TV market size, and income were statistically significant at the 5% level. In detail, a well-established total broadband infrastructure has a positive impact on OTT video streaming market growth. In addition, Netflix market entry is likely to increase paid OTT video streaming market growth. In particular, the time-lag model illustrates Netflix entry is likely to have an impact on paid OTT video streaming market growth, at least for one year after its entry. Furthermore, the traditional pay TV market size and the high income level are both positively associated with OTT market growth.

**Table 7.** Regressions of paid OTT video streaming market growth.

| Variable | Netflix (No Time-Lag) | | Netflix (1- and 2-Year Time-Lag) | |
|---|---|---|---|---|
| | coef B | *p*-Value | coef B | *p*-Value |
| ln_Total broadband infrastructure | 1.0054 | 0.000 | 0.8013 | 0.000 |
| OTT platform competition | 0.0000 | 0.992 | 0.00005 | 0.076 |
| Population density | −0.0004 | 0.831 | 0.0025 | 0.073 |
| Netflix market entry($t$) | 0.2642 | 0.030 | – | – |
| Netflix market entry($t$-1) | – | – | 0.3327 | 0.005 |
| Netflix market entry($t$-2) | – | – | 0.1599 | 0.195 |
| Traditional pay TV market size | 0.0003 | 0.002 | 0.0003 | 0.004 |
| Income | 0.0001 | 0.000 | 0.00003 | 0.000 |
| Constant | −4.3940 | 0.000 | −3.6451 | 0.000 |
| Country impact controlled | yes | | yes | |
| $F/\text{R}^2$ | $F(6,150) = 43.03/\text{R}^2 = 0.5748$ | | $F(7,143) = 34.39/\text{R}^2 = 0.6273$ | |
| Number of observations | 247 | | 200 | |

Table 8 presents the results of the regression analysis of the SVOD market growth model, controlling for country fixed effects. The F statistics in all regression models (both the no-time-lag model of SVOD market growth and the time-lag model) were statistically significant at the 5% level ($F(6,191) = 49.66$, $F(7,143) = 31.95$, respectively), which means that one of the regression coefficients would be different from zero. In the no-time-lag model, total broadband infrastructure, paid OTT market competition, and income were statistically significant at the 5% level. That is, with a well-constructed total broadband infrastructure, the SVOD market size is likely to expand. In addition, a high level of paid OTT platform competition is likely to lead to a larger SVOD market size, while high income level is positively associated with the SVOD market size. Though the no-time-lag model does not show the impact of Netflix entry, the time-lag model supported that Netflix entry is likely to contribute to the expansion of SVOD market size for one year after its entry, but not two years. Moreover, the model newly showed that traditional pay TV market size is positively associated with SVOD market growth.

**Table 8.** Regressions of SVOD market growth.

| Variable | Netflix (No Time-Lag) | | Netflix (1- and 2-Year Time-Lag) | |
|---|---|---|---|---|
| | coef B | *p*-Value | coef B | *p*-Value |
| ln_Total broadband infrastructure | 1.6336 | 0.000 | 0.68022 | 0.000 |
| OTT platform competition | −0.0008 | 0.000 | −0.00008 | 0.116 |
| Population density | 0.0001 | 0.968 | 0.0044 | 0.042 |
| Netflix market entry($t$) | 0.1252 | 0.684 | – | – |
| Netflix market entry($t$-1) | – | – | 0.7841 | 0.000 |
| Netflix market entry($t$-2) | – | – | 0.1523 | 0.424 |
| Traditional pay TV market size | 0.0004 | 0.144 | 0.0004 | 0.002 |
| Income | 0.0001 | 0.000 | 0.00007 | 0.000 |
| Constant | −5.803 | 0.054 | −6.248 | 0.000 |
| Country impact controlled | yes | | yes | |
| $F/R^2$ | $F(6,191) = 49.66/R^2 = 0.6094$ | | $F(7,143) = 31.95/ R^2 = 0.6100$ | |
| Number of observations | 247 | | 200 | |

Table 9 illustrates the results of the regression analysis of the pay TV subscription revenue growth rate, controlling for country effects. The F statistics in all regression models (both the overall and reduced model) were statistically significant at the 5% level ($F(6,191) = 49.66$, $F(5,151) = 14.93$, respectively), which means that one of the regression coefficients would be different from zero. The regression results provided that paid OTT video streaming revenue growth rate, PayTV/OTT platform market concentration, and Netflix market entry were statistically significant at the 5% level. Interestingly, as the OTT video streaming revenue growth rate increases, the pay TV subscription revenue growth rate is also likely to increase. In addition, PayTV/OTT platform market concentration is positively associated with an increase in the pay TV subscription revenue growth rate. The result also suggests that Netflix's entrance to the video market tends to reduce the pay TV subscription revenue growth rate. The reduced model supported these results in Table 9.

**Table 9.** Regressions of pay TV subscription revenue growth rate.

| Variable | Overall | | Reduced | |
|---|---|---|---|---|
| | coef B | *p*-Value | coef B | *p*-Value |
| ln_Paid OTT revenue growth rate | 0.0545 | 0.014 | 0.0544 | 0.013 |
| PayTV/OTT platform market concentration | 0.0002 | 0.005 | 0.00018 | 0.005 |
| Netflix market entry (*t*) | −0.1776 | 0.006 | −0.1772 | 0.006 |
| Income | −0.0000006 | 0.947 | −0.0000005 | 0.950 |
| Population density | 0.00004 | 0.963 | – | – |
| ln_fixed broadband infrastructure | −0.1804 | 0.292 | −0.181 | 0.287 |
| Constant | −3.0747 | 0.004 | −3.0555 | 0.002 |
| Country impact controlled | yes | | yes | |
| *F*/R$^2$ | $F_{(6,150)} = 12.36$/R$^2$= 0.3308 | | $F_{(5,151)} = 14.93$/R$^2$ = 0.3308 | |
| Number of observations | 203 | | 203 | |

## 5. Discussion and Conclusions

Employing panel data analysis, this study examined factors influencing early paid OTT video streaming market growth in 50 countries. One of the primary goals of this study was to analyze whether Netflix market entry, OTT platform competition, pay TV market size, and broadband infrastructure contribute to the market growth of paid OTT video streaming services.

We found support for RQ2. The results of the data analyses suggest that Netflix's market entry contributed to paid OTT video market growth. The results are consistent in the SVOD market growth model as well as in the paid OTT video streaming market growth model. It seems, in many countries, the initial effects of Netflix's market entry contributed to the growth of paid OTT video streaming markets. As Bourreau and Dogan [25] suggested, when a new firm enters the market, incumbent firms, such as cable and IPTV providers, improve their services through OTT strategies to compete with the innovative OTT firms. For instance, in order to keep their subscribers, some cable and IPTV providers offer platform-in-platform OTT services through their pay TV services. In addition, as innovative firms, such as Netflix, begin to take their market share, incumbent firms, such as telecommunication service providers and cable operators, employ diverse business strategies as a defense, which leads to their OTT market growth. It appears that the "catfish effect" has existed in the paid OTT video markets across many countries. RQ5 could be confirmed only for SVOD market growth. The statistical significance of the OTT platform competition variable in the SVOD market growth model suggests that strong platform competition between paid OTT platforms contributed to the market growth of SVOD services.

In addition, we found support for RQ1. Considering that the OTT revenue growth rate is positively associated with the pay TV subscription revenue growth rate, it appears that in initial media markets, OTT services and traditional pay TV services were growing together in many countries, which implies the relationship between paid OTT services and traditional pay TV services is complementary. We also found support for RQ3. The negative association between Netflix's market entry and the subscription revenue growth rate of traditional pay TV services implies that market entry and the growth of global OTT

video services, such as Netflix's streaming service, can potentially substitute the market growth of traditional pay subscription services, such as cable TV and IPTV, in the long run. Currently, traditional regulatory frameworks do not seem to be imposed on global OTT service providers, such as Netflix, Amazon Prime, and Disney Plus, in many countries. If OTT video services are significantly substituted for traditional pay TV subscription services in the long run, it may encourage discussion of regulatory frameworks for OTT video services in many countries for sustainable development of the media industry. Future studies need to examine the diverse effects of OTT service market growth on traditional broadcasting markets in the long run.

The findings here also suggest that a traditional pay TV market size is positively associated with OTT video market growth. Therefore, RQ7 could be confirmed. It seems that OTT video providers could expect higher potential returns in larger existing pay TV markets, where many users already have experience with purchasing video services. The findings also indicate that traditional pay TV subscription markets and OTT markets are growing together in many countries.

In addition, the results suggest that high levels of broadband infrastructure are associated with high levels of OTT video market growth. Therefore, RQ4 could be confirmed. This finding concurs with existing literature on the essential role of broadband infrastructures in ICT service diffusion and market growth. The results indicate that innovative ICT and media services can be well-developed under advanced broadband network conditions [36]. For instance, to offer high levels of real-time TV and diverse streaming video services through OTT platforms, mobile broadband infrastructures such as 4G and 5G are essential.

Furthermore, the statistical significance of concentration among pay TV platforms and OTT platforms indicates that high levels of platform market concentration among traditional pay TV platforms and paid OTT platforms contribute to high levels of pay TV subscription revenue growth. Therefore, RQ6 could be confirmed. Considering that traditional pay TV platforms currently have a larger market share than OTT platforms in many countries, this result is understandable.

This study has some limitations. For the data analyses, a relatively small number of observations was used given the relatively early market growth of OTT video. In addition, a sufficient number of OTT subscribers and OTT price data as well as traditional pay TV service price data were not available for the data analyses. By incorporating OTT subscribers and price data, future studies may investigate whether OTT video services are a substitute for traditional pay TV services. In addition, because of data availability issue, this study could not use very recent data for the analyses. Therefore, relatively old data were employed for the data analysis. If the future study employs recent OTT market data, the very recent market trend could be explained.

**Author Contributions:** Conceptualization, S.L. (Sangwon Lee) and S.L. (Seonmi Lee); methodology, S.L. (Sangwon Lee); formal analysis, S.L. (Sangwon Lee); data curation, H.J.; writing—original draft preparation, S.L. (Seonmi Lee); writing—review and editing, S.L. (Sangwon Lee) and Y.N.; project administration, Y.N.; funding acquisition, S.L. (Sangwon Lee). All authors have read and agreed to the published version of the manuscript.

**Funding:** This research was funded by KYUNG HEE UNIVERSITY, grant number [20181288].

**Conflicts of Interest:** The authors declare no conflict of interest.

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
