# Peer review of "Examining Factors Influencing Early Paid Over-The-Top Video Streaming Market Growth: A Cross-Country Empirical Study†"

_sustainability, doi:10.3390/su13105702_

Round 1

Reviewer 1 Report

The design of the research is fine. Data and variables have been carefully selected, explained and presented, including control and lagged variables. Some detailed remarks:

  • Figure 1 – Stacked histograms would be better and you will have the same information without three time series
  • The interpretation of F statistic in regression models is wrong. This test does not test the overall significance of the model. Null hypothesis (Ho) states that all regression coefficients equal zero, and alternative hypothesis (H1) states that at least one (!!!) regression coefficient is significantly different from zero. You can easily imagine the model with one significant variable, and 10 non-significant – with F statistic (correctly!) advocated to reject null, what does not mean that the model can be fairly interpreted.
  • I suggest not to report t statistics in Tables with regression results (reader should compare them with 1.96), but p-values. And don’t be afraid of p=0.0000 notation
  • It would be better to apply backwards step-wise regression to eliminate insignificant variables (one at a time!)

Reviewer 2 Report

The main aim of the paper is to analyze whether Netflix market entry, OTT platform competition, pay TV market size, and broadband infrastructure contributes to the market growth of paid OTT video streaming services. Datasets for 50 countries are relevant for doing research. The conclusions are rather expected, but in this paper, it is important to confirm them by statistical methods.

In conclusion, it would be useful to table the research questions and research results (confirmation / non-confirmation) and to support the conclusions through discussion.

Reviewer 3 Report

This is a very interesting study, although performed on somewhat old data, especially considering dynamic changes in the analyzed market. It should be also named as one of the study limitations.

In the introductory part you mention existence of several studies, but there are no sources/references.

There are some language issued that need to be looked upon  (for example instead of "in regard" (p.4, l.150) should be "regarding" or "with regard".
